# Victimization of Traditional and Cyber Bullying During Childhood and Their Correlates Among Adult Gay and Bisexual Men in Taiwan: A Retrospective Study

**DOI:** 10.3390/ijerph16234634

**Published:** 2019-11-21

**Authors:** Chien-Chuan Wang, Ray C. Hsiao, Cheng-Fang Yen

**Affiliations:** 1Zuoying Branch of Kaohsiung Armed Forces General Hospital, Kaohsiung 81342, Taiwan; jcwang93@gmail.com; 2Graduate Institute of Medicine and Department of Psychiatry, School of Medicine, College of Medicine, Kaohsiung Medical University, Kaohsiung 80708, Taiwan; 3Department of Psychiatry and Behavioral Sciences, School of Medicine, University of Washington, Seattle, WA 98195, USA; rhsiao@u.washington.edu; 4Department of Psychiatry, Children’s Hospital and Regional Medical Center, Seattle, WA 98105, USA; 5Department of Psychiatry, Kaohsiung Medical University Hospital, Kaohsiung 80708, Taiwan

**Keywords:** bullying, sexual minority, sexual orientation, gender role nonconformity, family support

## Abstract

This study examined the associations of timing of sexual orientation developmental milestones, gender role nonconformity, and family-related factors with victimization of traditional and cyber sexuality-related bullying during childhood among gay and bisexual men in Taiwan, in addition to the moderating effects of family-related factors on these associations. A total of 500 homosexual or bisexual men aged between 20 and 25 years were recruited into this study. The associations of early identification of sexual orientation, early coming out, level of masculinity, parental education levels, and perceived family support with victimization of traditional and cyber sexuality-related bullying were evaluated. Early identification of sexual orientation, low self-rated masculinity, and low family support were significantly associated with victimization of traditional bullying. Moreover, low family support, early coming out, and traditional bullying victimization were significantly associated with victimization of cyber bullying. Family support did not moderate the associations of early identification of sexual orientation and low masculinity with victimization of traditional bullying or cyberbullying. The factors associated with victimization of traditional and cyber sexuality-related bullying should be considered when mental health and educational professionals develop prevention and intervention strategies to reduce sexuality-related bullying.

## 1. Introduction

Victimization of bullying is one of the most miserable experiences children and adolescents can have and may result in long-term adverse psychological and physical consequences [1]. A meta-analysis study revealed that sexual minority youths, including lesbian, gay, bisexual, transgender, and questioning (LGBTQ) youths, reported higher victimization of bullying rates than heterosexual peers [2]. A review study revealed that victimization of bullying is one of the major factors contributing to mental disorders, suicide, and deliberate self-harm in sexual minority people [3]. Longitudinal studies have similarly demonstrated that victimization of bullying predicts subsequent psychological distress in sexual minority adolescents [4,5]. The aforementioned study findings thus support the implementation of public policy initiatives that reduce bullying and prevent victimization-related effects on the health and well-being of sexual minority youths [6]. Identifying factors that increase the risk of being bullied may provide fundamental knowledge for use in the design, implementation, and evaluation of interventions aiming to reduce bullying of sexual minority youths [4]. Cyber bullying, a new form of bullying, and its associated factors in sexual minority youths have not yet been surveyed thoroughly.

The minority stress hypothesis provides a perspective for understanding the factors related to victimization of bullying in sexual minority individuals. The hypothesis contends that the stigma, prejudice, and discrimination experienced by sexual minority individuals create a hostile social environment that can lead to chronic stress and mental health problems [7]. Being in a sexual minority is itself a minority stressor for LGBTQ youths. One longitudinal study discovered that sexual-minority-specific victimization not only was more prevalent among sexual minority youths compared with heterosexual counterparts but also significantly mediated the effect of sexual minority status on depressive symptoms and suicidality [5]. However, not every sexual minority individual experiences sexuality-related bullying; thus, there may be unique sexual minority stressors that increase the risk of experiencing sexuality-related bullying.

The first aim of the present study was to determine whether the early timing of sexual orientation developmental milestones and gender role nonconformity are sexual minority stressors for sexual minority youths. The major developmental milestones for sexual minorities include first experiencing same-gender attractions, first engaging in sexual behavior with same-gender individuals, first identifying as a sexual minority, first disclosing a sexual minority identity to others, and first same-gender relationship [8,9]. Research has found that the early timing of sexual orientation developmental milestones was associated with negative mental health outcomes such as depression and anxiety among adult lesbians and gay men [10,11], in addition to being associated with homelessness in sexual minority adolescents [12]. The process of sexual minority development differs in that sexual minority individuals face stigma related to sexual minority orientation, which may not only negatively affect the process of forming a minority sexual orientation but also increase the risk of being discriminated by peers. The development of neurocognitive function [13] and social skills [14] is ongoing during early adolescence and is associated with the relatively ineffective coping strategies of individuals in this developmental phase; therefore, early timing of sexual orientation developmental milestones may increase the risk of difficulties in peer interaction in childhood and adolescence. Further study is necessary to determine whether early identification of homosexual or bisexual orientation and early come out are associated with the risk of sexuality-related victimization of bullying among sexual minority individuals. Moreover, a literature review reported that gender role nonconformity significantly increases the risk of experiencing sexuality-related bullying in both heterosexual and sexual minority populations [15]. Individuals are expected to assume the roles and characteristics associated with their respective biological sex [16]. Those who do not assume the expected roles and characteristics of the gender associated with their biological sex are considered to be gender-nonconforming [17]. Gender-nonconforming boys who are more feminine than other boys can be described as those who transgress social gender norms [17].

However, the role played by gender role nonconformity regarding cyber sexuality-related bullying of sexual minority youths has not been examined. Further study is required to determine whether gender role nonconformity plays different roles within traditional face-to-face bullying and cyber bullying.

The socioecological framework developed by Bronfenbrenner [18] provides a perspective for understanding the role of family characteristics in sexuality-related bullying and the buffering effect that family characteristics have between the association of sexuality and gender role characteristics with bullying of sexual minority youths. Homophobic bullying is an ecological phenomenon according to the ecological systems perspective, and such bullying has been established and perpetuated over time as a result of the complex interactions between inter- and intraindividual factors [19]. Family characteristics may have a role for sexuality-related bullying in sexual minority individuals. For example, a study on the United States found that family-level microaggressions increases the risk of polyvictimization for sexual and gender minority adolescents [20].

Research has found that victimization of bullying in young people can be affected by family environment interactions such as domestic violence [21], low parental monitoring [22], and low parental warmth and family cohesion [23]. Given that a low parental educational level is associated with domestic violence and low family monitoring [24], one might hypothesize that young people with a low parental education level would have a higher risk of involvement in bullying than those with a high parental education level. Research found that a higher parental educational status is a protective factor for victimization of bullying in children but not in adolescents [25]. Therefore, the roles of family support and parental education level for victimization of sex-related traditional and cyber bullying warrant further study. Moreover, further study is necessary to determine the moderating effect of family support on the association of early timing of sexual orientation developmental milestones and gender role nonconformity with sexuality-related bullying of sexual minority youths.

People in East Asia are less tolerant of homosexuality than people in the Middle East and Africa; compared with European and North American countries, however, Asian countries generally exhibit much less tolerance of sexual minorities [26]. Due to the Confucian emphasis on family and kinship, homosexuality is regarded by East Asians as a challenge to the family obligations mandated in Confucianism, particularly to the requirement to continue the family bloodline, which explains the low tolerance to homosexuality in East Asian societies [26]. A study on sexual minority men in Japan revealed that 83% and 60% of the men experienced sexuality-related bullying, which increases the risk of attempted suicide [27]. Analysis of a national cross-sectional survey in Chinese adolescents found that victimization of bullying mediated the associations of sexual minority status with suicidality [28] and poor sleep quality [29]. Analysis of the Korea Youth Risk Behavior Web-based Survey found that same- and both-sexes intercourse related suicidality is strongly linked to victimization of bullying among youths [30]. Therefore, factors associated with sexuality-related victimization of bullying during childhood and adolescence in Asian countries warrant further investigation to offer a basis for developing prevention and intervention programs aimed at reducing bullying of sexual minority youths.

The aims of the present study were to examine whether there were differences in the timing of sexual orientation developmental milestones, gender role nonconformity, and family-related factors between gay and bisexual victims and non-victims of traditional and cyber sexuality-related bullying during childhood in Taiwan, in addition to the moderating effects that family factors have on the association of early timing of sexual orientation developmental milestones and gender role nonconformity with being victims of sexuality-related bullying. We had two research hypotheses. First, gay and bisexual victims of traditional and cyber sexuality-related bullying were more likely to identify their homosexual or bisexual orientation earlier, come out earlier, self-rate a lower level of masculinity, perceive lower family support, and have a lower parental education level than non-victims. Second, family support and parental education level moderated the relationships of early timing of sexual orientation developmental milestones and gender role nonconformity with being victims of sexuality-related bullying (Figure 1).

## 2. Methods

### 2.1. Participants

Participants were recruited using an online advertisement posted on Facebook, a bulletin board system, and the home pages of five health promotion and counseling centers for gay, lesbian, bisexual, and transgender (LGBT) individuals from August 2015 to July 2017. Print versions of the advertisement were also mailed to the LGBT student clubs of 25 colleges. A master-degree research assistant explained the study aims and procedures to potential participants who were interested in this study face-to-face and excluded two potential participants (one with impaired intellect and one with the smell of alcohol) who had difficulties in understanding the study’s purpose or and method to complete the questionnaire. In total, 500 participants (371 homosexual and 129 bisexual men) were recruited into this study. The mean age of the participants was 22.9 years (standard deviation (SD): 1.6 years). The sample size was calculated based on a previous study in Taiwan with the prevalence of traditional bullying 8.4% [31]. The estimated sample size was 426 with 80% power, 95% confidence interval (CI), and statistically significant level (α) at 5% [32]. The sample of 500 participants was thus determined as adequate. Informed consent was obtained from all participants prior to the assessment. The study was approved by the Institutional Review Board of Kaohsiung Medical University Hospital.

### 2.2. Measures

#### 2.2.1. Chinese Version of the School Bullying Experience Questionnaire

We used six items from the Chinese version of the self-report School Bullying Experience Questionnaire (C-SBEQ) to evaluate the participants’ experience of traditional sexuality-related victimization of bullying due to gender nonconformity and sexual orientation at school, in afterschool classes, at tutoring schools, and at part-time workplaces while they were a primary school and junior and senior high school student [33]. Two types of traditional bullying were surveyed, namely verbal ridicule and relational exclusion (three items: Social exclusion, being called a mean nickname, and being spoken ill of; for example, “How often have others spoken ill of you because they thought of you as a sissy [they found you homosexual or bisexual] in childhood or adolescence?”) and physical aggression and theft of belongings (three items: Being beaten up, being forced to do work, and having money, school supplies, or snacks taken away; for example, “How often have others beaten you up because they thought of you as a sissy [they found you homosexual or bisexual] in childhood or adolescence?”) These six items were rated on a 4-point Likert scale with 0 indicating *never*, 1 indicating *just a little*, 2 indicating *often*, and 3 indicating *all the time*. The psychometrics of the C-SBEQ have been examined elsewhere, and the results show that the C-SBEQ has good reliability and validity [33]. The total McDonald’s ω values of the scales for measuring the two types of victimization of traditional bullying due to gender nonconformity and sexual orientation were 0.85 and 0.92, respectively. According to the original study, participants who answered 2 or 3 to any item were identified as self-reported victims of traditional bullying [16].

#### 2.2.2. Cyberbullying Experiences Questionnaire

We employed three items from the Cyberbullying Experiences Questionnaire to assess the participants’ experience of cyber sexuality-related bullying victimization due to gender nonconformity and sexual orientation while a primary school and junior and senior high school student [34]. The three items addressed experience of mean or hurtful comments being posted about the participant; pictures, photos, or videos being posted that upset someone; and the spreading of rumors online through e-mails, blogs, social media (Facebook/Twitter/Plurk), pictures, or videos; for example, “How often have other students posted mean or hurtful comments on you through emails, blogs, or social media because they thought of you as a sissy (they found you homosexual or bisexual) in childhood or adolescence?” The items were rated on a 4-point Likert scale with 0 indicating *never*, 1 indicating *just a little*, 2 indicating *often*, and 3 indicating *all the time*. The total McDonald’s ω values of the scales for measuring victimization of cyber bullying due to gender nonconformity and sexual orientation were 0.76 and 0.80, respectively. According to the original study, participants who answered 1 or higher to any item were identified as self-reported victims of cyber bullying [34].

#### 2.2.3. Timing of Sexual Orientation Developmental Milestones and Gender Role Nonconformity

We collected the participants’ sexual orientation (homosexual or bisexual), age of identification of sexual orientation, and timing of coming out. Those who came out while in junior high school or earlier were classified as having come out early, whereas those who came out while in senior high school or after were classified as having come out late. We also evaluated the participants’ self-rated level of masculinity during childhood and adolescence using one item and a 5-point Likert scale ranging from 1 (*very low*) to 5 (*very high*).

We collected the participants’ sexual orientation (homosexual or bisexual), age of identification of sexual orientation (*“When did you firstly identify yourself as a gay or bisexual?”*), and timing of coming out (*“When did you firstly disclose your sexual identity to others?”*). Those who came out while in junior high school or earlier were classified as having come out early, whereas those who came out while in senior high school or after were classified as having come out late. We also evaluated the participants’ self-rated level of masculinity using one item (“Compared to other boys who are your same age, do you see yourself during childhood and adolescence as: Much more feminine (1), more feminine (2), about the same (3), more masculine (4), or much more masculine (5)?”) [17].

#### 2.2.4. Family-Related Factors

We examined the participants’ parental education levels and perceived family support during childhood and adolescence. In Taiwan, the duration of compulsory fundamental education is nine years. The participants were divided into those who had a high paternal and maternal education level (completing nine years of compulsory fundamental education) and those who had a low paternal and maternal education level (completing less than nine years of compulsory fundamental education). We employed the 5-item Chinese version of the Family Adaptation, Partnership, Growth, Affection, Resolve Index (APGAR) to measure the participants’ perceived family support using a 4-point Likert scale ranging from 0 (*never*) to 3 (*always*) [35,36]. High total scores indicate the perception of favorable family support. The total McDonald’s ω of APGAR in this study was 0.90.

### 2.3. Procedure and Data Analysis

A master-degree research assistant was responsible for administrating the research questionnaire after completing the training program. The questionnaire was administrated in the interview rooms of the research center that the principal investigator (CFY) worked at. The research assistant explained to the participants that the aims of this questionnaire-surveyed study was to explore the prevalence and risk factors of victimization of homophobic bullying among gay and bisexual men in Taiwan, and the results of this study might provide knowledge for use in the design, implementation, and evaluation of interventions aiming to reduce bullying of sexual minority youths. Then, the research assistant explained to the participants individually how to complete the questionnaires. The participants could ask questions when they encountered problems completing the questionnaires, and the research assistants would resolve their problems. The average time that the data collection process took overall was 30 min. Data analysis was performed using the SPSS 20.0 statistical software (SPSS Inc., Chicago, IL, USA).

The proportions of participants with experience of victimization of traditional and cyber sexuality-related bullying due to gender nonconformity and sexual orientation were calculated. The factors associated with traditional and cyber victimization of bullying were examined using two steps. First, differences in age, family characteristics, timing of sexual orientation developmental milestones, and level of masculinity between victims and nonvictims of traditional and cyber bullying were examined using chi-square and *t*-tests. A *p*-value of 0.05 or lower was used to indicate significance. The significant factors were then entered into multiple logistic regression analysis. Odds ratios (ORs) and 95% CIs were used to indicate statistical significance.

We also used the standard criteria proposed by Baron and Kenny [37] to examine the moderating effects of family-related factors on the association of early timing of sexual orientation developmental milestones and gender role nonconformity with sexuality-related bullying. According to these criteria, moderation occurred when the term representing interaction between the predictor (early timing of sexual orientation developmental milestones and gender role nonconformity) and the hypothesized moderator (family-related factors) was significantly associated with the dependent variable (sexuality-related bullying) after we controlled for the main effects of both the predictors and hypothesized moderator variables. If early timing of sexual orientation developmental milestones, gender role nonconformity, and family-related factors were significantly associated with sexuality-related bullying, the interactions (early timing of sexual orientation developmental milestones × family-related factors or gender role nonconformity × family-related factors) were incorporated into the regression analysis to examine the moderating effects.

## 3. Results

All 500 participants completed the research questionnaire without omission. The age, family characteristics, timing of sexual orientation developmental milestones, level of masculinity, and rates of victimization of traditional and cyber sexuality-related bullying of the 500 participants are presented in Table 1. In total, 23% and 22.4% of participants had a low paternal maternal education level, respectively. The mean (SD) of perceived family support on the APGAR was 8.5 (3.8). Regarding timing of sexual orientation developmental milestones, the mean (SD) of age to firstly identify sexual orientation was 13.8 (3.6) years old; 27.2% came out early. The mean (SD) level of self-rated masculinity was 2.7 (0.8). Regarding victimization of sexuality-related bullying during their childhood and adolescence, 34.8% and 17% reported to be victims of traditional bullying due to gender non-conformity and sexual orientation, respectively; 27% and 22.4% reported to be victims of cyber bullying due to gender non-conformity and sexual orientation, respectively. In total, 190 (38%) and 163 (32.6) participants reported themselves to be victims of traditional and cyber sexuality-related bullying, respectively.

The differences in age, family characteristics, sexual orientation developmental milestones, and the level of masculinity between the victims and non-victims of traditional and cyber bullying due to gender role nonconformity or sexual orientation during childhood and adolescence, obtained using chi-square and *t*-tests, are displayed in Table 2 and Table 3. Victims of traditional bullying had lower paternal and maternal education levels, perceived lower family support, were more likely to be gays, identified their sexual orientation earlier, came out earlier, and self-rated lower masculinity than non-victims of traditional bullying (Table 2).

Moreover, victims of cyber bullying perceived lower family support, came out earlier, and were more likely to be the victims of traditional bullying than non-victims of cyber bullying (Table 3).

The significant correlates of victimization of traditional bullying in chi-square and *t*-tests were entered into Model I of multiple logistic regression (Table 4). The results confirmed that lower family support, earlier identification of sexual orientation, and a lower level of masculinity were significantly associated with victimization of traditional bullying. The moderating effects of family support on the associations of early identification of sexual orientation and low masculinity with victimization of traditional bullying were further examined in Model II. The results of Model II revealed that neither the interaction variable of low family support × early identification of sexual orientation nor the interaction variable of low family support × low masculinity was significantly associated with victimization of traditional bullying, indicating that family support did not moderate the associations of early identification of sexual orientation and low masculinity with victimization of traditional bullying.

The significant correlates of victimization of cyber bullying in chi-square and *t*-tests were entered into multiple logistic regression models in two steps (Table 5). In the first step perceived family support and timing of coming out were selected into Model III. The results of Model III confirmed that low family support and early coming out were significantly associated with victimization of cyber bullying. In the second step, victimization of traditional bullying was further selected into Model IV. The results of Model IV revealed that victims of traditional bullying were more likely to also be victims of cyber bullying. The moderating effects of family support on the associations of early coming and victimization of traditional bullying with victimization of cyber bullying were further examined in Model V.

## 4. Discussion

The present study discovered that early identification of sexual orientation, low self-rated masculinity, and low family support were significantly associated with victimization of traditional bullying. Moreover, low family support, early coming out, and victimization of traditional bullying were significantly associated with victimization of cyber bullying. Family support did not moderate the associations of early identification of sexual orientation and low masculinity with victimization of traditional or cyber bullying.

The results of the present study supported the hypothesis that early identification of sexual orientation and early coming out are sexual minority stressors that may increase the risk of victimization of sexuality-related bullying. The developmental perspective may partially explain these results [11,38]. First, because of immature neurocognitive function and social skills, those who identify their sexual orientation or come out in early adolescence may be less able to cope effectively with stressors related to the stigma of sexual minority identification or to deal with bullying incidents compared with those who reach sexual orientation milestones in late adolescence or young adulthood. Research has found that early timing of sexual orientation developmental milestones was associated with negative mental health outcomes such as depression and anxiety among adult lesbians and gay men [10,11]. Second, individuals reaching sexual orientation developmental milestones earlier might have less access to supportive resources [11], which may increase their risk of being bullied. One study discovered that early timing of sexual orientation developmental milestones was significantly associated with homelessness among sexual minority adolescents [12]. The finding of the present study highlights the importance of developing strategies for the prevention and early detection of sexuality-related bullying of sexual minority youths who identify their sexual orientation or come out in early adolescence. Mental-health services providers and education professionals should provide sexual minority individuals who come out early the critical resources, including gay-straight alliances, inclusive curricular resources, supportive educators, and comprehensive bullying/harassment policies [39].

Low self-rated masculinity was significantly associated with victimization of traditional but not cyber bullying among sexual minority youths. Gender role conformity is a major component of heteronormativity, which prescribes gender norms and determines peer interactions [40]. One previous study found that endorsement of heteronormative culture or behavior contributes to the extent of homophobic bullying directed against sexual minority youths [41]. Boys who exhibit nonconformal gender characteristics such as low masculinity may be perceived as “gay” in face-to-face interactions and then be targets of homophobic bullying, whereas such characteristics may be considered less “odd” in cyber environments wherein gender role nonconformity is perceived or judged to a lesser extent. Sexual minority youths who exhibit obvious gender role nonconformity may perceive cyberspace as a safe environment compared with the face-to-face context and thus spend considerable time there. However, sexual minority youths are more likely to experience online peer victimization than heterosexual youths [42]. Moreover, cyber bullying is not easily detected by parents and school employees. Detection and intervention may not be possible until cyber bullying has severe consequences, especially in sexual minority youths without substantial gender role nonconformity.

The present study discovered that low family support was significantly associated with both victimization of traditional and cyber sexuality-related bullying of gay and bisexual youths. A similar result was obtained by a previous study; specifically, family acceptance reduced the effect of sexuality-based discrimination in gay and bisexual men [43]. There are several possible explanations for these results. First, low family support may make sexual minority youths feel uncomfortable about discussing with their families how to cope with the homophobic harassment they face. Repeated and prolonged harassment may then progress to homophobic bullying. Second, low family support may prevent youths from learning a mature and effective strategy from their parents that enables them to cope with the maltreatment from their peers. Third, sexual minority youths perceiving low family support may rely more heavily on peers than family members [44], and consequently, the risk of victimization of bullying increases. Fourth, low family support may co-occur with mental health problems, which may further increase the risk of being bullied. Contrary to our hypothesis, family support did not moderate the associations of early identification of sexual orientation and low masculinity with victimization of traditional or cyber bullying. Many sexual minority youths fear family rejection because of their sexual identity [45]. Thus, family buffering effects may be slight. The results of the present study indicate that interventions that help families become more accepting of young sexual minority family members may have beneficial mental health effects and reduce sexuality-related bullying.

Contrary to the hypothesis, this study did not find a significant association between low family support and victimization of sex-related traditional and cyber bullying in gay and bisexual men. Although research found that a higher parental educational status was a protective factor for victimization of bullying in children [25], the effect of parental education levels on adolescent victimization of bullying may abate as individuals grow up. Research found that most of people identify their sexual orientation during adolescence [46]. Therefore, the influence of parental educational level on victimization of sex-related bullying may be attenuated.

The present study found that victimization of traditional sexuality-related bullying increased the risk of victimization of cyber sexuality-related bullying among sexual minority youths. Previous studies have reported that cyberbullying perpetration is an extension of traditional bullying perpetration, particularly psychological, relational, and indirect forms of bullying in cyberspace [47,48]. Compared with traditional bullying perpetrators, cyberbullying perpetrators can remain virtually anonymous [49]. Being cruel and malicious using digital harassment is also easier because of the physical distance separating the offender and the victim [47]. All of these cyber activity characteristics may extend bullying perpetration from face-to-face interactional situations to cyberspace. The results of this study are a reminder to mental health and education professionals of the necessity of evaluating whether an individual is being bullied in the face-to-face context when addressing victimization of cyber bullying among sexual minority youths. Furthermore, because cyber bullying is less detectable by adults than traditional bullying is [50] and because most cyber bullying victims do not report such bullying to an adult or use digital tools to prevent online incidents [51], experiencing traditional bullying may be used as an indicator for detecting the occurrence of cyber bullying among sexual minority youths.

The present study is one of the first to examine the roles played by early timing of sexual orientation developmental milestones, gender role nonconformity, and family support regarding victimization of cyber sexuality-related bullying in sexual minority youths. The results of the present study also provided knowledges to the factors associated with sexuality-related victimization of bullying among sexual minority youths in Asian countries. However, the present study had several limitations. First, this study retrospectively obtained data on participants’ victimization of sexuality-related bullying, timing of sexual orientation developmental milestones, and family support; therefore, recall bias may have been introduced. Moreover, whether victimization of homophobic bullying in childhood and adolescence has adverse effects on victims’ memory in emerging adulthood warrants further study. Second, the study data were exclusively self-reported. The use of only a single data source may have influenced our findings and resulted in shared-method variances. Third, we did not examine the perpetrators of sexuality-related bullying. Fourth, the participants were gay or bisexual men who responded to the advertisements and participated in this study. Whether the results of this study can be generalized to those who did not respond to the advertisements warrants further study. Fifth, the cut-offs for identifying victims of traditional bullying (two or higher on the C-SBEQ) and cyber bullying (one or higher on the Cyberbullying Experiences Questionnaire) were not the same. Although the present study did not aim to compare the rates of victims between traditional bullying and cyber bullying, further study examining whether the relationship of traditional bullying with cyber bullying may vary if the cut-offs are changed may provide insights to the formation of cyber bullying.

Based on the results of the present study, we recommended further study to examine the mediators of the associations of early identification of sexual orientation, early come out, low masculinity, and low family support with victimization of traditional and cyber sexuality-related bullying. The identification of mediators not only provides knowledge to the occurrence of victimization of homophobic bullying but also serves as the target of prevention and intervention programs. Further prospective study is also needed to establish the temporal relationships among victimization of bullying and related individual and environmental factors, especially the relationship between victimization of traditional bullying and cyber bullying. Moreover, what kinds of cyberspace and cyber activities in which cyber homophobic bullying may occur also warrants further study. Sexual minority may experience not only homophobic bullying but also bullying related to other identity minority, for example, ethnicity, gender, and religion. Further study is needed to examine the experiences of victimization in double or multiple identity minority in Taiwan. Although parental educational levels were not significantly associated in the victimization of homophobic bullying in the present study, we were concerned that there may be other family factors, for example, parent-child bonding and parental knowledge and attitude toward sexual minority that relate to victimization of homophobic bullying in LGB individuals. We suggest further study to examine.

## 5. Conclusions

Based on the results of the present study, we suggest that factors associated with victimization of traditional and cyber sexuality-related bullying be considered when mental health and educational professionals develop a comprehensive approach to providing a positive school and community climate and reducing sexuality-related bullying [52]. Sexual minority youths who identify their sexual orientation and come out early could receive additional peer support from gay–straight alliances to reduce prejudice, discrimination, and bullying within schools [53]. Interventions that enhance family support for sexual minority youths may have beneficial mental health effects and reduce sexuality-related bullying. Whether sexual minority victims of cyber bullying also experience traditional bullying warrants routine surveying.

## Figures and Tables

**Figure 1 ijerph-16-04634-f001:**
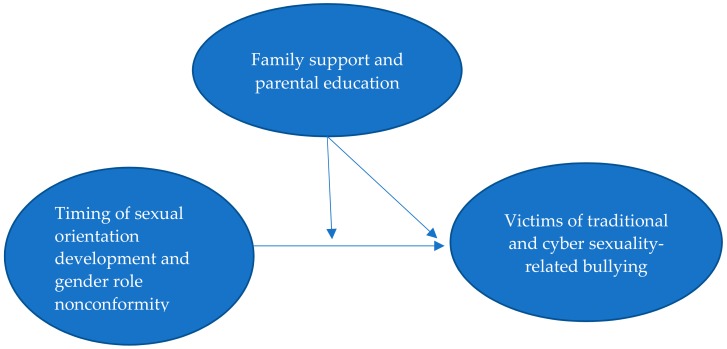
Hypothesized model of the associations among timing of sexual orientation development, gender role nonconformity, family-related factors, and victimization of traditional and cyber sexuality-related bullying.

**Table 1 ijerph-16-04634-t001:** Age, family characteristics, sexual orientation developmental milestones, level of masculinity, and traditional and cyber bullying victimization (*n* = 500).

Variables	*n* (%)	Mean (SD)	Range
Age (years)		22.9 (1.6)	20–25
Paternal education level			
High	385 (77)		
Low	115 (23)		
Maternal education level			
High	388 (77.6)		
Low	112 (22.4)		
Perceived family support on the APGAR		8.5 (3.8)	0–15
Sexual orientation			
Bisexuality	129 (25.8)		
Homosexuality	371 (74.2)		
Age of identification of sexual orientation (years)		13.8 (3.6)	6–23
Timing of coming out			
Late (senior high school or after)	364 (72.8)		
Early (junior high school or before)	136 (27.2)		
Self-rated level of masculinity		2.7 (0.8)	1–5
Victims of traditional bullying			
Due to gender non-conformity	174 (34.8)		
Due to sexual orientation	85 (17)		
Either	190 (38)		
Victims of cyber bullying			
Due to gender non-conformity	135 (27)		
Due to sexual orientation	112 (22.4)		
Either	163 (32.6)		

**Table 2 ijerph-16-04634-t002:** Correlates of victimization of traditional bullying during childhood and adolescence: Chi-square and *t*-tests (*n* = 500).

Variables	Traditional Bullying				
	No (*n* = 310)	Yes (*n* = 190)	χ^2^ or *t*	*p*	Cohen’s d
Paternal education level, *n* (%)					
High (*n* = 385)	250 (64.9)	135 (35.1)	6.121	0.013	
Low (*n* = 115)	60 (52.2)	55 (47.8)			
Maternal education level, *n* (%)					
High (*n* = 388)	250 (64.4)	138 (35.6)	4.352	0.037	
Low (*n* = 112)	60 (53.6)	52 (46.4)			
Perceived family support, mean (SD)	9.1 (3.6)	7.5 (4.0)	4.809	<.001	0.44
Sexual orientation, *n* (%)					
Bisexuality (*n* = 129)	95 (73.6)	34 (26.4)	10.004	0.002	
Homosexuality (*n* = 371)	215 (58.0)	156 (42.0)			
Age of identification of sexual orientation (years), mean (SD)	14.3 (3.4)	13.0 (3.7)	4.099	<0.001	0.37
Timing of coming out, *n* (%)					
Late (senior high school or after) (*n* = 364)	237 (65.1)	127 (34.9)	5.493	0.019	
Early (junior high school or before) (*n* = 136)	73 (53.7)	63 (46.3)			
Self-rated level of masculinity, mean (SD)	2.9 (0.8)	2.3 (0.8)	7.882	<0.001	0.72
Victims of traditional bullying, *n* (%)					
No (*n* = 310)					
Yes (*n* = 190)					

**Table 3 ijerph-16-04634-t003:** Correlates of victimization of cyber bullying during childhood and adolescence: Χ^2^ and *t-* tests (*n* = 500).

Variables	Cyber Bullying				
	No (*n* = 299)	Yes (*n* = 201)	χ^2^ or *t*	*p*	Cohen’s d
Paternal education level, *n* (%)					
High (*n* = 385)	229 (59.5)	156 (40.5)	0.071	0.790	
Low (*n* = 115)	70 (60.9)	45 (39.1)			
Maternal education level, *n* (%)					
High (*n* = 388)	237 (61.1)	151 (38.9)	1.185	0.276	
Low (*n* = 112)	62 (55.4)	50 (44.6)			
Perceived family support, mean (SD)	8.9 (3.8)	7.8 (3.8)	3.148	0.002	0.29
Sexual orientation, *n* (%)					
Bisexuality (*n* = 129)	75 (58.1)	54 (41.9)	0.199	0.655	
Homosexuality (*n* = 371)	224 (60.4)	147 (39.6)			
Age of identification of sexual orientation (years), mean (SD)	14.1 (3.5)	13.5 (3.7)	1.779	0.076	
Timing of coming out, *n* (%)					
Late (senior high school or after) (*n* = 364)	231 (63.5)	133 (36.5)	7.463	0.006	
Early (junior high school or before) (*n* = 136)	68 (50)	68 (50)			
Self-rated level of masculinity, mean (SD)	2.7 (0.8)	2.6 (0.9)	1.374	0.170	
Victims of traditional bullying, *n* (%)					
No (*n* = 310)	218 (70.3)	92 (29.7)	35.575	<0.001	
Yes (*n* = 190)	81 (42.6)	109 (57.4)			

**Table 4 ijerph-16-04634-t004:** Correlates of victimization of traditional bullying during childhood and adolescence: Multiple logistic regression (*n* = 500).

Variables	Victims of Traditional Bullying			
	Model I		Model II	
	OR	95% CI of OR	OR	95% CI of OR
Low paternal education	1.439	0.859–2.411	1.444	0.861–2.421
Low maternal education	1.084	0.646–1.822	1.085	0.646–1.823
Perceived family support	0.894	0.847–0.943	0.843	0.647–1.098
Homosexuality (bisexuality as reference)	1.262	0.768–2.074	1.270	0.772–2.091
Age of identification of sexual orientation	0.939	0.884–0.998	0.912	0.794–1.046
Early coming out	1.122	0.704–1.788	1.122	0.703–1.791
Self-rated level of masculinity	0.401	0.302–0.532	0.388	0.206–0.734
Perceived family support × Age of identification of sexual orientation			1.004	0.989–1.019
Perceived family support × Level of masculinity			1.004	0.932–1.082
−2 log likelihood	570.916		570.672	
Nagelkerke R^2^	0.231		0.232	
Walds χ^2^	28.232		28.232	
*p*	<0.001		<0.001	

**Table 5 ijerph-16-04634-t005:** Correlates of victimization of cyber bullying during childhood and adolescence: Multiple logistic regression (*n* = 500).

Variables	Victims of Cyber Bullying					
	Model III		Model IV		Model V	
	OR	95% CI of OR	OR	95% CI of OR	OR	95% CI of OR
Perceived family support	0.928	0.885–0.974	0.952	0.906–1.001	0.865–1.002	0.865–1.002
Early coming out	1.729	1.156–2.584	1.585	1.045–2.403	0.391–2.875	0.391–2.875
Victims of traditional bullying			2.868	1.949–4.221	1.012–6.199	1.012–6.199
Perceived family support × Early coming out					0.921–1.126	0.921–1.126
Perceived family support × Victims of traditional bullying					0.942–1.170	0.942–1.170
−2 log likelihood	656.919		627.830		626.891	
Nagelkerke R^2^	0.045		0.119		0.121	
Walds χ^2^	18.958		18.958		18.958	
*p*	<0.001		<0.001		<.001	

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
