# Peer review of "Victimization of Traditional and Cyber Bullying During Childhood and Their Correlates Among Adult Gay and Bisexual Men in Taiwan: A Retrospective Study"

_ijerph, 2019, doi:10.3390/ijerph16234634_

Round 1

Reviewer 1 Report

This study investigated the traditional- and cybervictimization of LGBTQ youths in relation to timing of sexual orientation developmental milestones, gender role nonconformity and family–related factors. This study is meaningful in that there are very limited studies on Asian LGBTQ youths’ mental health. The background of this study needs more explanation.

Term use: “Bullying victimization” has been used as one term in this study, however it means different aspects of one phenomenon. It may be more appropriate to use on term, either bullying or victimization. Personally, this study is about victimization rather than bullying behavior thus, it may also be good to use a term ‘victimization’ (including the title of the study).

Introduction:

This study indicates early timing of sexual orientation developmental milestones can be a stressor for victimization; the reason why early coming out is more likely than late coming out to be at risk should be explained in Introduction (It was explained partly in discussion, but in introduction,

I am not an expert on LGBTQ studies, so it would be helpful if the clear meaning of ‘sexual orientation developmental milestones’ is added

Also, why family characteristics including parental education level and family supports are important protective factors need to be more added in introduction; Furthermore, parental education levels were hardly considered in discussion

The strengths of this study which distinguished from previous literature should be explained.

Discussion:

The suggestions for helping youth coming out early should be included. Also, victims in traditional bullying were related to victims in cyberbullying (57.4%)(Table 2). The possible relation victim experience between in TB and CB among LGBTQ youths could be suggested.

Hope my comments can be helpful for revising the manuscript.

Author Response

Comment

This study investigated the traditional- and cybervictimization of LGBTQ youths in relation to timing of sexual orientation developmental milestones, gender role nonconformity and family–related factors. This study is meaningful in that there are very limited studies on Asian LGBTQ youths’ mental health. The background of this study needs more explanation.

Response

Thank you for your suggestion. In the revised manuscript we added more explanations for background of Asian LGBTQ youths’ mental health as below. Please refer to line 112-126.

“People in East Asia are less tolerant of homosexuality than people in the Middle East and Africa; compared with European and North American countries, however, Asian countries generally exhibit much less tolerance of sexual minorities [26]. Because of the Confucian emphasis on family and kinship, homosexuality is regarded by East Asians as a challenge to the family obligations mandated in Confucianism, particularly to the requirement to continue the family bloodline, which explains the low tolerance to homosexuality in East Asian societies [26]. A study on sexual minority men in Japan revealed that 83% and 60% of the men experienced sexuality-related bullying, which increases the risk of attempted suicide [27]. Analysis of a national cross-sectional survey in Chinese adolescents found that victimization of bullying mediated the associations of sexual minority status with suicidality [28] and poor sleep quality [29]. Analysis of the Korea Youth Risk Behavior Web-based Survey found that same- and both-sexes intercourse related suicidality is strongly linked to victimization of bullying among youths [30]. Therefore, factors associated with sexuality-related victimization of bullying during childhood and adolescence in Asian countries warrant further investigation to offer a basis for developing prevention and intervention programs aimed at reducing bullying of sexual minority youths.”

Comment

Term use: “Bullying victimization” has been used as one term in this study, however it means different aspects of one phenomenon. It may be more appropriate to use on term, either bullying or victimization. Personally, this study is about victimization rather than bullying behavior thus, it may also be good to use a term ‘victimization’ (including the title of the study).

Response

We revised the term “bullying victimization” into “victimization of bullying” thorough the revised manuscript.

Comment

This study indicates early timing of sexual orientation developmental milestones can be a stressor for victimization; the reason why early coming out is more likely than late coming out to be at risk should be explained in Introduction (It was explained partly in discussion, but in introduction.)

Response

Thank you for your comment. In Introduction section of the revised manuscript we added the possible reason why early coming out is more likely than late coming out to be at risk as below. Please refer to line 66-78.

“Research has found that the early timing of sexual orientation developmental milestones was associated with negative mental health outcomes such as depression and anxiety among adult lesbians and gay men [10, 11], in addition to being associated with homelessness in sexual minority adolescents [12]. The process of sexual minority development differs in that sexual minority individuals face stigma related to sexual minority orientation, which may not only negatively affect the process of forming a minority sexual orientation but also increase the risk of being discriminated by peers. The development of neurocognitive function [13] and social skills [14] is ongoing during early adolescence and is associated with the relatively ineffective coping strategies of individuals in this developmental phase; therefore, early timing of sexual orientation developmental milestones may increase the risk of difficulties in peer interaction in childhood and adolescence. Further study is necessary to determine whether early identification of homosexual or bisexual orientation and early come out are associated with the risk of sexuality-related victimization of bullying among sexual minority individuals.”

Comment

I am not an expert on LGBTQ studies, so it would be helpful if the clear meaning of ‘sexual orientation developmental milestones’ is added.

Response

We added the meaning of ‘sexual orientation developmental milestones for sexual minority into Introduction section. Please refer to line 63-66.

“The major developmental milestones for sexual minorities include first experiencing same-gender attractions, first engaging in sexual behavior with same-gender individuals, first identifying as a sexual minority, first disclosing a sexual minority identity to others, and first same-gender relationship [8, 9].”

Comment

Why family characteristics including parental education level and family supports are important protective factors need to be more added in introduction. Parental education levels were hardly considered in discussion.

Response

We added explanations for the reason to examine the protective effects of family support and parental education level on victimization of sex-related traditional and cyber bullying in the revised manuscript. Please refer to line 100-108.

“Research has found that victimization of bullying in young people can be affected by family environment interactions such as domestic violence [21], low parental monitoring [22], and low parental warmth and family cohesion [23]. Given that a low parental educational level is associated with domestic violence and low family monitoring [24], one might hypothesize that young people with a low parental education level would have a higher risk of involvement in bullying than those with a high parental education level. Research found that a higher parental educational status is a protective factor for victimization of bullying in children but not in adolescents [25]. Therefore, the roles of family support and parental education level for victimization of sex-related traditional and cyber bullying warrant further study.”

In the revised manuscript we also added discussion regarding the role of parental education level as below. Please refer to line 373-379.

“Contrary to the hypothesis, this study did not find a significant association between low family support and victimization of sex-related traditional and cyber bullying in gay and bisexual men. Although research found that a higher parental educational status was a protective factor for victimization of bullying in children [25], the effect of parental education levels on adolescent victimization of bullying may abate as individuals grow up. Research found that most of people identify their sexual orientation during adolescence [46]. Therefore, the influence of parental educational level on victimization of sex-related bullying may be attenuated.”

Comment

The strengths of this study which distinguished from previous literature should be explained.

Response

Thank you for your suggestion. We added the strengths of the present study as below into the Discussion section. Please refer to line 395-399.

“The present study is one of the first to examine the roles played by early timing of sexual orientation developmental milestones, gender role nonconformity and family support regarding victimization of cyber sexuality-related bullying in sexual minority youths. The results of the present study also provided knowledges to the factors associated with sexuality-related victimization of bullying among sexual minority youths in Asian countries.”

Comment

The suggestions for helping youth coming out early should be included. Also, victims in traditional bullying were related to victims in cyberbullying (57.4%) (Table 2). The possible relation victim experience between in TB and CB among LGBTQ youths could be suggested.

Response

Thank you for suggestion. We included the suggestion for helping youth coming out early in Discussion section. Please refer to line 336-339.

“Mental-health services providers and education professionals should provide sexual minority individuals who come out early the critical resources, including gay-straight alliances, inclusive curricular resources, supportive educators, and comprehensive bullying/harassment policies [39].”

We also added a paragraph in Discussion section to discuss the relationship between traditional bullying and cyber bulling. Please refer to line 381-388.

“Previous studies have reported that cyberbullying perpetration is an extension of traditional bullying perpetration, particularly psychological, relational, and indirect forms of bullying in cyberspace [47, 48]. Compared with traditional bullying perpetrators, cyberbullying perpetrators can remain virtually anonymous [49]. Being cruel and malicious using digital harassment is also easier because of the physical distance separating the offender and the victim [47]. All of these cyber activity characteristics may extend bullying perpetration from face-to-face interactional situations to cyberspace.”

Reviewer 2 Report

INTRODUCTION

Obviously, the authors handle a large number of variables, so they must improve the formulation of objectives. The hypotheses must also be clearly formulated, where the variables must appear expressly. Therefore, it is recommended to present the hypotheses separately and identify in some way (for example, with numbers). This will facilitate the reading and monitoring of results and the subsequent discussion.

On the other hand, considering that the topic is currently changing, perhaps you should review the citations in the introduction section and update some of them or include more recent ones (more representative of the current state of the matter).

It would also be helpful, and since relationships between multiple variables are analyzed, that the authors represent this in a figure, where the formulated hypotheses are included graphically.

METHOD

The description of the characteristics of the sample should be in the section of participants, not results.

As Cronbach’s α has some deficiencies (i.e., relying on assumptions that are very strict and unrealistic), I strongly recommend to calculate McDonald’s ω (McDonald, 1999) as a measure of internal consistency that makes fewer and more realistic assumptions (Dunn et al., 2014). Alternatively, you could test if the assumption of Cronbach’s α – the essentially tau-equivalent measurement model – is fulfilled (Graham, 2006).

RESULTS

A lot of relevant information are missing. Please see for instance American Psychological Association (2013) for reporting standards.

In this section it would also be appropriate to present a figure that shows the relationships between the variables.

DISCUSSION

Emphasize the limitations of the study, for example, in relation to the selection of participants.

Author Response

Comment

Obviously, the authors handle a large number of variables, so they must improve the formulation of objectives. The hypotheses must also be clearly formulated, where the variables must appear expressly. Therefore, it is recommended to present the hypotheses separately and identify in some way (for example, with numbers). This will facilitate the reading and monitoring of results and the subsequent discussion.

Response

Thank you for your suggestion. We revised the formulation of objectives and hypotheses as below. Please refer to line 127-138.

“The aims of the present study were to examine whether there were differences in timing of sexual orientation developmental milestones, gender role nonconformity, and family-related factors between gay and bisexual victims and non-victims of traditional and cyber sexuality-related bullying during childhood in Taiwan, in addition to the moderating effects that family factors have on the association of early timing of sexual orientation developmental milestones and gender role nonconformity with being victims of sexuality-related bullying. We had two research hypotheses. First, gay and bisexual victims of traditional and cyber sexuality-related bullying were more likely to identify their homosexual or bisexual orientation earlier, come out earlier, self-rate a lower level of masculinity, perceive lower family support, and have a lower parental education level than non-victims. Second, family support and parental education level moderated the relationships of early timing of sexual orientation developmental milestones and gender role nonconformity with being victims of sexuality-related bullying (Figure 1).”

Comment

On the other hand, considering that the topic is currently changing, perhaps you should review the citations in the introduction section and update some of them or include more recent ones (more representative of the current state of the matter).

Response

Thank you for your suggestion. In the revised manuscript we added four citations published between 2016-2019 in Introduction, mainly the studies on East Asian countries that have attitudes toward sexual minority similar to Taiwan. Please refer to line 96-99 and line 119-126.

Comment

It would also be helpful, and since relationships between multiple variables are analyzed, that the authors represent this in a figure, where the formulated hypotheses are included graphically.

Response

Thank you for your suggestion. We added Figure 1 to illustrated the hypothesized model. Please refer to line 140-144.

Comment

The description of the characteristics of the sample should be in the section of participants, not results.

Response

We moved the description of the characteristics of the sample to the section of participants. Please refer to line 155-157.

Comment

As Cronbach’s α has some deficiencies (i.e., relying on assumptions that are very strict and unrealistic), I strongly recommend to calculate McDonald’s ω (McDonald, 1999) as a measure of internal consistency that makes fewer and more realistic assumptions (Dunn et al., 2014). Alternatively, you could test if the assumption of Cronbach’s α – the essentially tau-equivalent measurement model – is fulfilled (Graham, 2006).

Response

Thank you for your suggestion. We calculated the total McDonald’s ω values as the measure of internal consistency in the revised manuscript. Please refer to line 179-181, line 193-195 and line 222.

“The total McDonald’s ω values of the scales for measuring the two types of victimization of traditional bullying due to gender nonconformity and sexual orientation were .85 and .92, respectively.” “The total McDonald’s ω values of the scales for measuring victimization of cyber bullying due to gender nonconformity and sexual orientation were .76 and .80, respectively.” “The total McDonald’s ω of APGAR in this study was 0.90.”

Comment

A lot of relevant information are missing. Please see for instance American Psychological Association (2013) for reporting standards.

Response

Thank you for your reminding. We added the relevant information based on the reporting standards proposed by American Psychological Association (2013) in the revised manuscript, including:

Date defining the periods of recruitment (“from August 2015 to July 2017”). Please refer to line 150. Missing data (“All 500 participants completed the research questionnaire without omission.) Please refer to line 258. measures that characterize the data employed (“In total, 23% and 22.4% of participants had a low paternal maternal education level, respectively. The mean (SD) of perceived family support on the APGAR was 8.5 (3.8). Regarding timing of sexual orientation developmental milestones, the mean (SD) of age to firstly identify sexual orientation was 13.8 (3.6) years old; 27.2% came out early. The mean (SD) level of self-rated masculinity was 2.7 (0.8). Regarding victimization of sexuality-related bullying during their childhood and adolescence, 34.8% and 17% reported to be victims of traditional bullying due to gender non-conformity and sexual orientation, respectively; 27% and 22.4% reported to be victims of cyber bullying due to gender non-conformity and sexual orientation, respectively. In total, 190 (38%) and 163 (32.6) participants reported themselves to be victims of traditional and cyber sexuality-related bullying, respectively.”). Please refer to line 261-270.

Comment

In this section it would also be appropriate to present a figure that shows the relationships between the variables.

Response

We added Figure 1 to illustrated the hypothesized model to present a figure that shows the relationships between the variables. Please refer to line 140-144.

Comment

Emphasize the limitations of the study, for example, in relation to the selection of participants.

Response

Thank you for your reminding. In the revised manuscript we listed the selection of participants as one of limitations of this study as below. Please refer to line 407-409.

“The participants were gay or bisexual men who responded to the advertisements and participated into this study. Whether the results of this study can be generalized to those who did not respond to the advertisements warrants further study.”

Reviewer 3 Report

Thank you for sharing the manuscript for peer review. The study is interesting but I have some concerns related with the research design and methodology. Those concerns should be solved before recomending acceptance.

1) The retrospective nature of the study should be described in the introduction and even in the title described that the gay and bisexual men participating are adults.

2) The introduction is well-built and is supported on a theoretical framework. However, other retrospective studies should be reviewed in order to justify the importance of conducting retrospective studies even when there are inherent methodological problems when we measure variables experienced in the past.

3) Participants. Although is is described later in the manuscript, please include mean age and sexual orientation distribution in the participants description.

4) Authors stated that "individuals who exhibited any deficits...were excluded". Please, explain how this information was obtained in order to fully understand the selection procedure.

5) Authors should include the exact question preceding the bullying and cyberbullying forms. I mean...how participants were asked about their childhood experiences with bullying and cyberbullying.

6) It seems problematic that the bullying and cyberbullying scales had a different Likert-type scale. This is one of the limitations of the study and authors should explain how they solved this because the categorization procedure for bullying and cyberbullying victims was also different given the differences in the Likert-type scale. Please, also include the explanation all of the options for the cyberbullying questionnaire, as was included for the bullying scale.

7) It is also problematic that masculinity was only measured by one item given that different participants may have different ideas of what masculinity is. Was given a masculinity definition to the participants? Was is related with traditional gender roles or traditional gender traits? Please, justify this for of measuring masculinity.

8) Procedure information is too limited. How the study was explained to the participants? Did the authors run a power analysis for the selection?Where were the questionnaires administered? How long did the data collection process take overall?

9) Results: t-test and chi-squared test allow as to test differences between two groups. When authors described their results write about associations. I think it is more appropriated to write about the differences found and write about associations in the multiple logistic regression analyses.

10) Provide Cohen’s d in all the significant t-tests.

11) Logistic regression analyses are difficult to follow. Please, explain in detail what variables where introduce in each model. Moreover, I do not understand why interaction variables are not included in the table and why interaction results are explained before that the results of the initial models.

12) Authors should include  statistical data like (-2 LL), Nagelkerke R2, Value of the Model X² in the logistic regressions in order to understand the significance of the results.

13) Please, mark significance on the variables (table 3) that reach significance in the regression analysis.

14) Discussion is well drawn on the results obtained, but I would like to see more elaboration about future research recommendations.

I wish all the luck in revising your paper.

Author Response

Comment

1) The retrospective nature of the study should be described in the introduction and even in the title described that the gay and bisexual men participating are adults.

Response

Thank you for your suggestion. We revised the title to be “Victimization of Traditional and Cyber Bullying During Childhood and Their Correlates Among Adult Gay and Bisexual Men in Taiwan: A Retrospective Study.” Please refer to line 2-5.

Comment

2) The introduction is well-built and is supported on a theoretical framework. However, other retrospective studies should be reviewed in order to justify the importance of conducting retrospective studies even when there are inherent methodological problems when we measure variables experienced in the past.

Response

Thank you for your suggestion. In the revised manuscript we added the results of four retrospective or cross-sectional studies published between 2016-2019 in Introduction, mainly the studies on East Asian countries that have attitudes toward sexual minority similar to Taiwan. Please refer to line 96-99 and line 119-126.

Comment

3) Participants. Although it is described later in the manuscript, please include mean age and sexual orientation distribution in the participants description.

Response

Thank you for your suggestion. We moved the description of mean age and sexual orientation distribution of the sample to the section of participants. Please refer to line 155-157.

Comment

4) Authors stated that "individuals who exhibited any deficits...were excluded". Please, explain how this information was obtained in order to fully understand the selection procedure.

Response

Thank you for your suggestion. We added explanation for the selection procedure as below. Please refer to line 151-156.

“A master-degree research assistant explained the study aims and procedures to potential participants who were interested in this study face-to-face and excluded two potential participants (one with impaired intellect and one with the smell of alcohol) who had difficulties in understanding the study’s purpose or and method to complete the questionnaire.”

Comment

5) Authors should include the exact question preceding the bullying and cyberbullying forms. I mean...how participants were asked about their childhood experiences with bullying and cyberbullying.

Response

Thank you for your suggestion. We added explanation for the selection procedure as below.

“How often have others spoken ill of you because they thought of you as a sissy [they found you homosexual or bisexual] in childhood or adolescence?” Please refer to line 171-173.

“How often have others beaten you up because they thought of you as a sissy [they found you homosexual or bisexual] in childhood or adolescence?” Please refer to line 174-176.

“How often have other students posted mean or hurtful comments on you through emails, blogs, or social media because they thought of you as a sissy (they found you homosexual or bisexual) in childhood or adolescence?” Please refer to line 190-192.

Comment

6) It seems problematic that the bullying and cyberbullying scales had a different Likert-type scale. This is one of the limitations of the study and authors should explain how they solved this because the categorization procedure for bullying and cyberbullying victims was also different given the differences in the Likert-type scale. Please, also include the explanation all of the options for the cyberbullying questionnaire, as was included for the bullying scale.

Response

Thank you for your reminding. Although we chose the cut-offs according to the original studies, we added it as one of points that need further study. Please refer to line 409-414.

“The cut-offs for identifying victims of traditional bullying (2 or higher on the C-SBEQ) and cyber bullying (1 or higher on the Cyberbullying Experiences Questionnaire) were not the same. Although the present study did not aim to compare the rates of victims between traditional bullying and cyber bullying, further study examining whether the relationship of traditional bullying with cyber bullying may vary if the cut-offs are changed may provide insights to the formation of cyber bullying.”

We added the options for the cyberbullying questionnaire as below. Please refer to line 192-193.

“The items were rated on a 4-point Likert scale with 0 indicating never, 1 indicating just a little, 2 indicating often, and 3 indicating all the time. “

Comment

7) It is also problematic that masculinity was only measured by one item given that different participants may have different ideas of what masculinity is. Was given a masculinity definition to the participants? Was is related with traditional gender roles or traditional gender traits? Please, justify this for of measuring masculinity.

Response

Thank you for your comment. We added a short paragraph do introduce the concepts of gender role nonconformity and masculinity/femininity as below in Introduction section of the revised manuscript. Please refer to line 81-85.

“Individuals are expected to assume the roles and characteristics associated with their respective biological sex [16]. Those who do not assume the expected roles and characteristics of the gender associated with their biological sex are consider to be gender-nonconforming [17]. Gender-nonconforming boys who are more feminine than other boys can be described as those who transgress social gender norms [17].”

We also added the content of the item inquiring the self-rated level of masculinity adopted from the study of Toomey and colleagues (2010) into the revised manuscript. Please refer to line 209-212.

“We also evaluated the participants’ self-rated level of masculinity using one item (“Compared to other boys who are your same age, do you see yourself during childhood and adolescence as: Much more feminine (1), more feminine (2), about the same (3), more masculine (4), or much more masculine (5)?”) [17].”

Comment

8) Procedure information is too limited. How the study was explained to the participants? Did the authors run a power analysis for the selection? Where were the questionnaires administered? How long did the data collection process take overall?

Response

Thank you for your comments. In the revised manuscript we added introduction about the procedure of survey, including how to explain this study to the participants, where to administrate the questionnaire, and how long the data collection process took overall. Please refer to line 224-234.

“A master-degree research assistant was responsible for administrating the research questionnaire after completing the training program. The questionnaire was administrated in the interview rooms of the research center that the principal investigator (CFY) worked at. The research assistant explained to the participants that the aims of this questionnaire-surveyed study was to explore the prevalence and risk factors of victimization of homophobic bullying among gay and bisexual men in Taiwan, and the results of this study might provide knowledge for use in the design, implementation, and evaluation of interventions aiming to reduce bullying of sexual minority youths. Then the research assistant explained to the participants individually how to complete the questionnaires. The participants could ask questions when they encountered problems completing the questionnaires, and the research assistants would resolve their problems. The average time that the data collection process took overall was 30 minutes.”

We did run a power analysis for the number of participants. We added it (as below) into the revised manuscript. Please refer to line 157-160.

“The sample size was calculated based on a previous study in Taiwan with the prevalence of traditional bullying 8.4 percent [31]. The estimated sample size was 426 with 80 percent power, 95 percent confidence interval (CI), and statistically significant level (α) at 5 percent [32]. The sample of 500 participants was thus determined as adequate.”

Comment

9) Results: t-test and chi-squared test allow as to test differences between two groups. When authors described their results write about associations. I think it is more appropriated to write about the differences found and write about associations in the multiple logistic regression analyses.

Response

Thank you for your comment. In the revised manuscript we rewrote the paragraph describing the results of t-test and chi-squared test listed in Table 2 as below. Please refer to line 276-279 and line 283-285.

“Victims of traditional bullying had lower paternal and maternal education levels, perceived lower family support, were more likely to be gays, identified their sexual orientation earlier, came out earlier, and self-rated lower masculinity than non-victims of traditional bullying (Table 2)... Moreover, victims of cyber bullying perceived lower family support, came out earlier, and were more likely to be the victims of traditional bullying than non-victims of cyber bullying (Table 3).”

Comment

10) Provide Cohen’s d in all the significant t-tests.

Response

We added Cohen’s d in all the significant t-tests into Table 2 and Table 3. Please refer to line 280 and line 286.

Comment

11)

Logistic regression analyses are difficult to follow. Please, explain in detail what variables where introduce in each model. Moreover, I do not understand why interaction variables are not included in the table and why interaction results are explained before that the results of the initial models.

Response

We rewrote the paragraph describing the processes of multiple logistic regression following the sequence of Model I to Model V as below.

“The significant correlates of victimization of traditional bullying in chi-square and t tests were entered into Model I of multiple logistic regression (Table 4)... The moderating effects of family support on the associations of early identification of sexual orientation and low masculinity with victimization of traditional bullying were further examined in Model II.” Please refer to line 289-294.

“The significant correlates of victimization of cyber bullying in chi-square and t tests were entered into multiple logistic regression models in two steps (Table 5). In the first step perceived family support and timing of coming out were selected into Model III… In the second step victimization of traditional bullying was further selected into Model IV...The moderating effects of family support on the associations of early coming and victimization of traditional bullying with victimization of cyber bullying were further examined in Model V.” Please refer to line 303-311.

In the revised manuscript we included the interaction variables in Table 4 and Table 5. Please refer to line 300 and line 312. We also changed the sequence of initial models (Model I) and interaction results (Model II) for traditional bullying, as well as the sequence of initial models (Model III and Model IV) and interaction results (Model V) for cyber bullying. Please refer to line 292-294 and line 309-311.

Comment

12) Authors should include statistical data like (-2 LL), Nagelkerke R2, Value of the Model X² in the logistic regressions in order to understand the significance of the results.

Response

Thank you for your suggestion. We added the statistical data including -2 log likelihood, Nagelkerke R2 and value of the Model χ² into Table 4 and Table 5. Please refer to line 300 and line 312.

Comment

13) Please, mark significance on the variables (table 3) that reach significance in the regression analysis.

Response

We labeled the OR and 95% CI of the significant variables with bold numbers in Table 4 and Table 5. Please refer to line 300 and line 312.

Comment

14) I would like to see more elaboration about recommendations.

Response

Thank you for your suggestion. We added a new paragraph recommending future research as below. Please refer to line 415-431.

“Based on the results of the present study, we recommended further study to examine the mediators of the associations of early identification of sexual orientation, early come out, low masculinity and low family support with victimization of traditional and cyber sexuality-related bullying. The identification of mediators no only provides knowledge to the occurrence of victimization of homophobic bullying but also serves as the target of prevention and intervention programs. Further prospective study is also needed to establish the temporal relationships among victimization of bullying and related individual and environmental factors, especially the relationship between victimization of traditional bullying and cyber bullying. Moreover, what kinds of cyberspace and cyber activities in which cyber homophobic bullying may occur also warrants further study. Sexual minority may experience not only homophobic bullying but also bullying related to other identity minority, for example, ethnicity, gender, and religion. Further study is need to examine the experiences of victimization in double or multiple identity minority in Taiwan. Although parental educational levels were not significantly associated victimization of homophobic bullying in the present study, we concerned that there may be other family factors, for example, parent-child bonding and parental knowledge and attitude toward sexual minority that relate to victimization of homophobic bullying in LGB individuals. We suggest further study to examine.”

Reviewer 4 Report

The present study set out to examine the relation between early sexual orientation, gender nonconformity and family support with traditional and cyber bullying during childhood in gay and bisexual men in Taiwan. In this study authors have observed that low family support, early identification of sexual orientation is significantly associated with traditional and cyber bullying. They also observed that family support did not moderate the association of early sexual orientation with traditional and cyber bullying. 

To my opinion this is good observation and key finding also has sufficient references to support their observation.  The present study adds to the growing body of research that indicates factors associated with traditional and cyber bullying victimization should be considered when mental health and early life stress effects programmes are developed. 

In this regards I have few minor comments:

Did authors perform any cognition related test e.g. TMT-A or TMT-B on the study subjects to investigate effect of early childhood bullying on their memory? Did authors perform BDI test on subjects to study their depression behaviour? Was their any effect observe if one parent has low educational level and other parent has high education level in the family? Ethnic background information is not given in the methods section. Although study was performed in Taiwan Men but did authors observed same or mixed ethnic background among study subjects. Did author consider ethnic background information while analyzing the data?

Author Response

Comment

Did authors perform any cognition related test e.g. TMT-A or TMT-B on the study subjects to investigate effect of early childhood bullying on their memory?

Response

We did not perform any cognition related test. We agree that early childhood bullying may have adverse effects on victims’ memory and increase the possibility of recall bias in retrospective study. We added it as one of points warranting further study as below. Please refer to line 400-404.

“This study retrospectively obtained data on participants’ victimization of sexuality-related bullying, timing of sexual orientation developmental milestones, and family support; therefore, recall bias may have been introduced. Moreover, whether victimization of homophobic bullying in childhood and adolescence has adverse effects on victims’ memory in emerging adulthood warrants further study.”

Comment

Did authors perform BDI test on subjects to study their depression behaviour?

Response

We did evaluate participants’ depression using the Center for Epidemiological Studies-Depression Scale and examined the relationship between victimization of homophobic bullying with depression in gay and bisexual men. The results have been published (Wang et al., Neuropsychiatric Disease and Treatment 2018;14:1309–1317).

Comment

Was there any effect observe if one parent has low educational level and other parent has high education level in the family?

Response

We agree that there may be different effects on adolescent health between paternal and maternal educational levels. For example, our previous study on 10,233 adolescents in Taiwan found that low maternal but not paternal education level was significantly associated with adolescent suicidality (Tang et al., Suicide and Life-Threatening Behavior 2009; 39(1):91-102.). Although parental educational levels were not significantly associated victimization of homophobic bullying in the present study, we concerned that there may be other family factors, for example, parent-child bonding and parental knowledge and attitude toward sexual minority that relate to victimization of homophobic bullying in LGB individuals. We suggest further study to examine. We added it into the revised manuscript for further study. Please refer to line 427-431.

“Although parental educational levels were not significantly associated victimization of homophobic bullying in the present study, we concerned that there may be other family factors, for example, parent-child bonding and parental knowledge and attitude toward sexual minority that relate to victimization of homophobic bullying in LGB individuals. We suggest further study to examine.”

Comment

Ethnic background information is not given in the methods section. Although study was performed in Taiwan men but did authors observe same or mixed ethnic background among study subjects. Did author consider ethnic background information while analyzing the data? 

Response

The issue of victimization of homophobic bullying in sexual minority who are also ethic minority is definitely important because they may experience double stigma. The indigenous population in Taiwan consists of 16 different tribes, approximately 2% of the population of Taiwan. The present study did not examine ethnic background among participants. We recommended further study examining the experiences of victimization of both homophobic and ethic-related bullying in sexual and ethic minority in Taiwan. Please refer to line 424-427.

“Sexual minority may experience not only homophobic bullying but also bullying related to other identity minority, for example, ethnicity, gender, and religion. Further study is need to examine the experiences of victimization in double or multiple identity minority in Taiwan.”

Round 2

Reviewer 3 Report

Literature review has improved. Authors described now the distinctive features of their study and why the study is needed. Method is now better described to allow replicability. Statistic are now well-reported and the data is significant and important. In my opinion, the article merits publication.